# Microscopy and Image Analysis-Based Identification of High- and Low-Pressure Mechanically Separated Pork Meat

**DOI:** 10.3390/foods14244295

**Published:** 2025-12-13

**Authors:** Marzia Pezzolato, Elena Biasibetti, Francesco Pennisi, Giovanna Esposito, Riccardo Provera, Nunzia Giaccio, Cristiana Maurella, Elena Bozzetta

**Affiliations:** Istopatologia Department, Istituto Zooprofilattico Sperimentale del Piemonte, Liguria e Valle d’Aosta Via Bologna 148, 10154 Torino, Italy; marzia.pezzolato@izsplv.it (M.P.); francesco.pennisi@izsplv.it (F.P.); giovanna.esposito@izsplv.it (G.E.); riccardo.provera@izsplv.it (R.P.); nunzia.giaccio@izsplv.it (N.G.); cristiana.maurella@izsplv.it (C.M.); elena.bozzetta@izsplv.it (E.B.)

**Keywords:** mechanically separated pork meat, histology, image analysis

## Abstract

Mechanically separated meat (MSM) is a product of animal origin obtained by applying mechanical force to remove residual meat from bones, a process that results in the loss or modification of muscle fiber structure. Its use is regulated in the European Union. Clearly distinguishing MSM from non-MSM is essential not only to maintain high product standards but also to ensure full compliance with food labeling regulations and to meet consumer expectations for transparency. This study aimed to evaluate a standardized histological protocol for differentiating high-pressure MSM, low-pressure MSM, and non-MSM pork meat using light microscopy, combined with image analysis and statistical evaluation. In total 40 MSM samples (20 high-pressure, 20 low-pressure) and 20 non-MSM samples were analyzed. The histological protocol achieved 100% sensitivity and specificity in distinguishing MSM from non-MSM. Image analysis revealed significant differences between high- and low-pressure MSM in calcium aggregate parameters. The method provides a reliable, cost-effective tool for MSM identification in routine food inspection. However, because the study was conducted exclusively on pork samples, the results should be interpreted within the context of this type of meat. Additional validation across other species and production systems is required to confirm the broader applicability of the method.

## 1. Introduction

Mechanically separated meat (MSM) is a product of animal origin obtained by removing meat from flesh-bearing bones or poultry carcasses using mechanical means. This process results in the loss or modification of muscle fiber structure and produces a paste-like product. Its use in the European Union must comply with Regulation (EC) No 853/2004 [1], which imposes strict labeling requirements due to differences in composition, microbiological quality, and nutritional value compared to conventionally deboned meat. As specified in Annex I, MSM must be clearly declared in the ingredients list, indicating the species of origin, and products containing MSM are required to provide mandatory information for the final consumer, including a statement that they must be cooked before consumption [1].

Recent market data highlight the relevance of MSM in the European meat supply chain [2]. Although official statistics seldom specify MSM volumes, its use is widespread in processed meat products, particularly in low-cost formulations, due to its cost-effectiveness. Regulatory agencies, including the UK Food Standards Agency, have recently updated guidance on MSM labeling to improve compliance [3].

MSM production parameters, particularly the pressure applied during separation, can significantly influence muscle fiber integrity, bone content, and mineral composition [4]. These factors impact not only product classification but also shelf life and microbiological safety. EU-wide monitoring of *Escherichia coli* in MSM and meat preparations has been implemented to harmonize hygiene assessment [5]. The EFSA One Health 2023 Zoonoses Report highlighted MSM-related pathogen detections across member states, underscoring public health concerns [6]. High-pressure MSM has been associated with greater mechanical disruption of bone and cartilage, potentially influencing microbial load and physicochemical quality [7]. While MSM helps maximize carcass yield and reduce waste, there are concerns regarding non-compliance with labeling requirements, which may constitute a form of food fraud when MSM is used without proper declaration [8]. Traditional detection methods often rely on calcium quantification; however, this approach can be insufficient due to variability in raw materials and processing [9]. The EFSA BIOHAZ Panel has therefore recommended the development of alternative or complementary methods for MSM identification [10]. For this reason, histology combined with Von Kossa staining was selected, as it offers high specificity for visualizing calcium deposits and outperforms stains such as Masson’s trichrome, Alcian Blue, or Alizarin Red in detecting mineralized particles. Recent technological advances offer promising solutions. Non-thermal treatments such as high-pressure processing (HPP), ultrasound, ozone, organic acids, ultraviolet light, and cold plasma can extend MSM shelf life and reduce bacterial contamination [11]. Specifically, HPP is increasingly used in the meat industry, representing approx. 25–30% of the processing volume, and offers benefits for microbial inactivation, texture, and water-holding capacity [12,13]. Analytical advances include hyperspectral imaging coupled with deep learning models, capable of identifying tissue contaminants on pork products with high accuracy [14], and micro-computed tomography (micro-CT) imaging for detecting residual bone particles [15]. Nevertheless, these technologies often require specialized equipment and expertise, thus excluding their adoption for routine regulatory settings.

In this context, the paper aims to highlight how histological analysis, especially when combined with Von Kossa staining for calcium deposits and quantitative image analysis, represents a cost-effective, widely accessible tool for distinguishing MSM from non-MSM and for assessing processing conditions such as applied pressure. This approach aligns with EFSA recommendations [10] and offers practical advantages for routine food inspection and fraud prevention.

## 2. Materials and Methods

### 2.1. Sample Collection

A total of 40 reference MSM pork samples (20 high-pressure, 20 low-pressure) were collected from two authorized Italian production plants. Additionally, 20 fresh ground pork (non-MSM) samples served as negative controls.

### 2.2. Histological Preparation

Samples were fixed in 10% neutral buffered formalin, routinely processed and paraffin-embedded, sectioned (3 ± 2 μm), and stained with hematoxylin–eosin. Two histological slides were prepared from each sample (80 MSM slides, 40 control slides). Von Kossa staining was also performed on each sample to visualize calcium deposits, with a bovine tuberculous lymph node as positive control.

### 2.3. Image Analysis

Ten random microscopic fields from each Von Kossa-stained slide were analyzed at 10× magnification using NIS-Elements AR 4.50.00 software (Nikon Europe B.V., Amstelveen, The Netherlands). Parameters considered included the total area occupied by calcium, the number of fields containing aggregates, the total aggregate count, and the mean number of aggregates per field.

### 2.4. Statistical Analysis

Data were analyzed using Kruskal–Wallis tests to compare high- vs. low-pressure MSM. Sensitivity, specificity, likelihood ratios (LR+ and LR−), and ROC curves were calculated for each parameter. Inter-rater agreement between two blinded pathologists was assessed using Cohen’s kappa. Statistical analysis was performed in Stata/SE 14.0 (StataCorp, College Station, TX, USA).

## 3. Results

### 3.1. Histological Differentiation of MSM vs. Non-MSM

All 20 non-MSM samples lacked bone and/or cartilage presence within tissues. Calcium deposits (colored in brown) were present in all MSM samples but absent in non-MSM. All 80 MSM samples were correctly identified due to the presence of bone and/or cartilage inside muscle, connective, and/or adipose tissues. (Figure 1) With Von Kossa staining, all calcium deposits appear reddish/brownish in color, while the background is uniformly light pink, much paler than with traditional hematoxylin–eosin staining. Sensitivity and specificity for distinguishing MSM from non-MSM were both 100% (95% CI: 95–100%), with a Cohen’s kappa of 1.00 indicating perfect inter-rater agreement.

### 3.2. Image Analysis and Statistical Comparison

Statistical analysis of data obtained by image analysis (see Table 1, Table 2, Table 3, Table 4, Table 5, Table 6, Table 7 and Table 8) revealed significant differences in calcium aggregate distribution, especially in high-pressure MSM samples. For instance, Table 2 and Table 3 demonstrate a significantly higher number of fields with aggregates in high-pressure MSM (*p* < 0.0001, ROC area = 1.000), while Table 5 and Table 7 highlight the diagnostic value of aggregate count and mean aggregates per field (ROC area = 0.9212 and 0.7288, respectively).

## 4. Discussion

The present study confirms that histological examination, supported by the calcium-specific Von Kossa staining, can distinguish MSM from conventionally deboned pork meat with perfect diagnostic accuracy. These findings are consistent with EFSA recommendations to develop novel microscopy-based approaches when traditional biochemical parameters, such as calcium quantification alone, are insufficient for regulatory purposes [10]. The method demonstrated high reproducibility, with perfect inter-rater agreement, and identified statistically significant differences in calcium aggregate distribution between high- and low-pressure MSM.

From a regulatory perspective, this approach is particularly remarkable in light of updated MSM guidance from the UK Food Standards Agency [3] and recent EU-wide monitoring efforts targeting *E. coli* in MSM and meat preparations [5]. The EFSA One Health 2023 Zoonoses Report [6] has highlighted the presence of MSM-related pathogens in the food chain, reinforcing the need for reliable, accessible, and cost-effective diagnostic tools to ensure compliance with Regulation (EC) No 853/2004 [1].

The differences observed between high- and low-pressure MSM likely reflect the greater mechanical disruption of bone and cartilage at higher pressures [7], which, while maximizing yield, may also cause increased bone tissue separation and can influence microbiological quality too. This aligns with literature that links the intensity of the process to increased microbial loads and potential quality deterioration in MSM [11]. However, the absence of a significant difference in the total area occupied by calcium between different pressure groups suggests that particle distribution metrics, rather than total mineral content, may offer better discriminatory power.

Compared with emerging technologies such as hyperspectral imaging coupled with deep learning [14], micro-computed tomography [15], and high-pressure processing (HPP) monitoring systems [12], histology offers the advantages of minimal equipment requirements, relatively short turnaround time, and compatibility with existing infrastructure in official control laboratories. While advanced methods provide additional molecular or three-dimensional structural information, their cost and technical complexity limit routine implementation in most inspection settings.

Finally, the method presented here also addresses food fraud prevention: undeclared MSM use in meat preparations not only constitutes a breach of labeling laws but may also mislead consumers regarding product quality [8]. By enabling both the detection of MSM and assessment of processing conditions, the proposed approach supports enforcement actions and consumer health protection.

## 5. Conclusions

This study validates a standardized histological protocol including Von Kossa staining, for the detection of MSM in pork meat, achieving 100% sensitivity and specificity. It also demonstrates the utility of quantitative image analysis for differentiating high-pressure from low-pressure MSM based on selected parameters. Although this approach requires validation in other species, the results obtained in pork meat highlight its robustness. The method aligns with EFSA and FSA recommendations for reliable and robust MSM detection [3,10], while offering a cost-effective and practical solution suitable for integration into routine official controls.

The approach holds exceptional value in contexts where advanced analytical technologies, such as LC–MS/MS proteomics, hyperspectral imaging, or micro-CT, are unavailable due to cost or technical constraints. By enabling rapid screening of meat products, it can help reduce the incidence of undeclared MSM, thereby supporting compliance with EU Regulation (EC) No 853/2004 and enhancing consumer trust.

## Figures and Tables

**Figure 1 foods-14-04295-f001:**
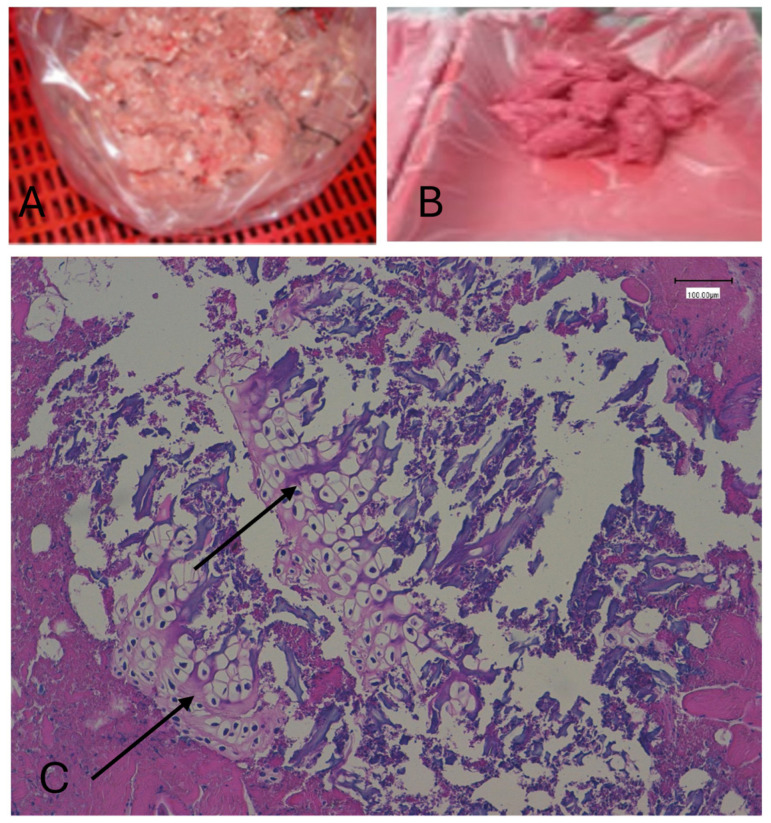
Low-pressure MSM (**A**); high-pressure MSM (**B**); hematoxylin–eosin-stained MSM showing bone/cartilage fragments. The black arrow indicates bone/cartilage (**C**).

**Table 1 foods-14-04295-t001:** Total calcium-occupied area.

	Obs	Rank Sum
HIGH PRESSURE	20	476.00
LOW PRESSURE	20	344.00

Kruskal–Wallis equality-of-populations rank test: χ^2^ = 3.187 with 1 d.f.; probability = 0.0742. No significant difference between high- and low-pressure MSM (*p* = 0.0742).

**Table 2 foods-14-04295-t002:** Number of fields containing aggregates.

	Obs	Rank Sum
HIGH PRESSURE	20	610.00
LOW PRESSURE	20	210.00

Kruskal–Wallis equality-of-populations rank test: χ^2^ = 29.268 with 1 d.f.; probability = 0.0001. Number of fields with aggregates significantly higher in high-pressure MSM (*p* < 0.0001).

**Table 3 foods-14-04295-t003:** Number of fields containing aggregates. Cutpoint data.

Cutpoint	Sensitivity	Specificity	Classified	LR+	LR−
(≥0)	100.00	0.00	50	1	
(≥1)	100.00	15.00	57.5	1.1765	0.0000
(≥2)	100.00	30.00	65.00	1.4286	0.0000
(≥3)	100.00	85.00	92.50	6.6667	0.0000
(≥4)	100.00	100.00	100.00		0.0000
(≥5)	85.00	100.00	92.50		0.1500
(≥6)	45.00	100.00	72.50		0.5500
(≥7)	30.00	100.00	65.00		0.7000
(≥8)	15.00	100.00	57.50		0.875
(≥10)	5.00	100.00	52.50		0.9500
(>10)	0.00	100.00	50.00		1.000

**Table 4 foods-14-04295-t004:** Number of fields containing aggregate: ROC area.

ROC				
Obs	Area	Std. Err.	[95% Conf. Interval]	
≥4.0	1.000	0.0000	1.0000	1.0000

ROC curves. Number of fields with aggregates significantly higher in high-pressure MSM (ROC area = 1.000).

**Table 5 foods-14-04295-t005:** Number of aggregates per slide.

	Obs	Rank Sum
HIGH PRESSURE	20	578.50
LOW PRESSURE	20	241.50

Significantly higher in high-pressure MSM Kruskal–Wallis equality-of-populations rank test: χ^2^ with ties = 20.828 with 1 d.f.; probability = 0.0001 (*p* < 0.0001).

**Table 6 foods-14-04295-t006:** Number of aggregates per slide: ROC area.

ROC				
Obs	Area	Std. Err.	[95% Conf. Interval]	
≥10	0.9212	0.0533	0.81679	1.0000

Significantly higher in high-pressure MSM; ROC area = 0.9212.

**Table 7 foods-14-04295-t007:** Mean aggregates per field.

	Obs	Rank Sum
HIGH PRESSURE	20	501.50
LOW PRESSURE	20	318.50

Significantly higher in high-pressure MSM. Kruskal–Wallis equality-of-populations rank test: χ^2^ with ties = 6.147 with 1 d.f. probability = 0.0132 (*p* = 0.0132).

**Table 8 foods-14-04295-t008:** Mean aggregates per field: ROC area.

ROC				
Obs	Area	Std. Err.	[95% Conf. Interval]	
≥1.833	0.7288	0.0869	0.55836	0.89914

Mean aggregates per field: Significantly higher in high-pressure (ROC area = 0.7288).

## Data Availability

The original contributions presented in this study are included in the article. Further inquiries can be directed to the corresponding author.

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
