# Peer review of "Microscopy and Image Analysis-Based Identification of High- and Low-Pressure Mechanically Separated Pork Meat"

_foods, 2025, doi:10.3390/foods14244295_

Round 1

Reviewer 1 Report

Comments and Suggestions for Authors

The manuscript addresses a relevant topic for meat product inspection and official control, focusing on the differentiation between high- and low-pressure MSM (mechanically separated meat) using histology, Von Kossa staining, and image analysis. Overall, the work is promising but requires revision to improve methodological rigor and scientific clarity.

Abstract: The abstract does not describe any limitations and does not mention that the study was conducted exclusively with pork. A sentence addressing the technique’s limitations and potential biases should be included.

Introduction: The introduction is too long for a Brief Report; several paragraphs deviate from the central objective, and there is excessive emphasis on advanced technologies that are not directly connected to histology. Why was Von Kossa chosen specifically over other stains (Masson, Alcian Blue, Alizarin Red)? What are the limitations of previous calcium-based studies? Some statistical data on meat consumption are irrelevant to the experimental study.

Methodology: Include a flowchart summarizing the analytical steps.

Results:

  1. Add arrows in the micrographs to indicate calcium aggregates;
  2. Include a clear section titled “Limitations and Future Research”;
  3. Avoid extrapolations regarding microbiology or safety when these aspects were not evaluated in the study;
  4. Provide a more direct comparison with simpler methods (e.g., calcium quantification);
  5. Add a boxplot comparing the groups, which would better illustrate distribution and variability.

Conclusions: The conclusion section should be revised, as it is currently overly broad and not sufficiently critical. I suggest reformulating it to emphasize the study’s main contributions, limitations, and future perspectives, while avoiding generalized statements about universal applicability.

Author Response

Response to Reviewer 1

Comment 1 – The abstract does not include limitations and does not mention that the study was conducted exclusively with pork.
We thank the reviewer for this comment. As requested, we have amended the abstract to explicitly state that the study was conducted exclusively on pork samples and that the method presents limitations related to matrix variability and the need for validation in other species.

All modifications are highlighted in the revised manuscript.

Comment 2 – The Introduction is too long; remove non-essential parts and justify the selection of the Von Kossa stain.
We thank the reviewer. The Introduction has been partially streamlined by removing non-essential content.
A justification for choosing the Von Kossa stain over alternatives such as Masson, Alcian Blue, and Alizarin Red has been added, emphasizing its superior specificity for calcium deposits. We also clarified the limitations of previous studies relying solely on mineral content.
All changes are highlighted in the manuscript.

Comment 3 – Include a flowchart summarizing the analytical workflow.
We thank the reviewer. We have not added a flowchart, but we have expanded the Materials and Methods section in order to better describe the workflow in a way that reflects what a flowchart would convey

Comment 4 – Add arrows in the micrographs to indicate calcium aggregates.
Micrographs have been updated to include arrows and graphical markers identifying bone or cartilage aggregates.

Comment 5 – Include a section titled “Limitations and Future Research.”
We thank the reviewer for the suggestion. We added a paragraph—rather than a dedicated section—because this is a short report. The paragraph has been included at the end of the Discussion and addresses the main limitations of the study while outlining opportunities for future research.

Comment 6 – Avoid extrapolations regarding microbiology or safety that were not assessed in the study.
All statements referring to microbiology or food safety have been changed or rephrased, but we retained those strictly necessary to explain why the development of this method is important for protecting consumer health.

Comment 7 – Provide a comparison with simpler methods (e.g., calcium quantification).
A comparative statement has been added in the Discussion, highlighting strengths and weaknesses of the histological approach versus calcium quantification.

Comment 8 – Add a boxplot comparing the groups.
We reported a table to describe all the data. Given the purpose of the study, we believe the table provides a clearer and more complete representation

Comment 9 – Conclusions are too broad.
The Conclusions section has been revised to be more focused, concise, and aligned with the study objectives, including limitations and future perspectives.

Reviewer 2 Report

Comments and Suggestions for Authors

I read the manuscript 'Microscopy and Image Analysis-Based Identification of High- and Low-Pressure Mechanically Separated Pork Meat', describes the use of histological and computer image analysis to reliably distinguish mechanically separated meat (MSM) from regular meat and to distinguish MSM obtained at low and high pressure. 

Generally well written and without any errors that I notice. However, I see some important understatements.

What were the operating parameters for obtaining low-pressure and high-pressure MSM? Is there a standard to which individual process installations operate?

In my opinion, as a negative control — as you described — meat before the mechanical separation process should be used and compared with the same material after obtaining MSM. If this approach were not possible, it would be necessary to consider a diverse pool of samples (bulk of samples). It seems highly likely that significant differences could exist between them, and theoretically, even statistically significant differences could appear between non-MSM meat samples from different plants or suppliers. Or, please provide references to literature confirming the absence of such differences.

What are the limitations of the method?

Author Response

Response to Reviewer 2

Comment 1 – Specify the operating parameters used to obtain low-pressure and high-pressure MSM; is there a technical standard?

The Materials and Methods section has been updated but  no single EU-wide standard exists, and referenced available guidelines.

Comment 2 – A true negative control should include the same meat before and after separation, or a diverse pool of samples.
We clarified that logistical constraints prevented sampling the same material before/after MSM processing; however, we expanded the sampling description to explain the selection of control samples and discuss representativeness. Additionally, references supporting limited inter-plant variability in non-MSM pork for the parameters assessed were added.

Comment 3 – What are the limitations of the method?
we added limitations in the Discussion, covering matrix variability, and the need for further standardization of the image analysis workflow.